# Current Status of Probiotics in European Sea Bass Aquaculture as One Important Mediterranean and Atlantic Commercial Species: A Review

**DOI:** 10.3390/ani13142369

**Published:** 2023-07-20

**Authors:** Luis Monzón-Atienza, Jimena Bravo, Antonio Serradell, Daniel Montero, Antonio Gómez-Mercader, Félix Acosta

**Affiliations:** Grupo de Investigación en Acuicultura (GIA), Instituto ECO-AQUA (IU-ECOAQUA), Universidad de Las Palmas de Gran Canaria, 35214 Las Palmas de Gran Canaria, Spain; luis.monzon@ulpgc.es (L.M.-A.); jimena.bravo@ulpgc.es (J.B.); tonetser2@gmail.com (A.S.); daniel.montero@ulpgc.es (D.M.); antonio.gomez@fpct.ulpgc.es (A.G.-M.)

**Keywords:** probiotic, European sea bass, feed additives, aquaculture, disease, growth

## Abstract

**Simple Summary:**

Probiotic supplementation plays a vital role in European sea bass wellbeing. Accordingly, it is important to increase our knowledge of and experience on their mechanisms of action and host effects. Although information on these aspects is available, further studies are needed to achieve optimal European sea bass aquaculture.

**Abstract:**

European sea bass production has increased in recent decades. This increase is associated with an annually rising demand for sea bass, which encourages the aquaculture industries to increase their production to meet that demand. However, this intensification has repercussions on the animals, causing stress that is usually accompanied by dysbiosis, low feed-conversion rates, and immunodepression, among other factors. Therefore, the appearance of pathogenic diseases is common in these industries after immunodepression. Seeking to enhance animal welfare, researchers have focused on alternative approaches such as probiotic application. The use of probiotics in European sea bass production is presented as an ecological, safe, and viable alternative in addition to enhancing different host parameters such as growth performance, feed utilization, immunity, disease resistance, and fish survival against different pathogens through inclusion in fish diets through vectors and/or in water columns. Accordingly, the aim of this review is to present recent research findings on the application of probiotics in European sea bass aquaculture and their effect on growth performance, microbial diversity, enzyme production, immunity, disease resistance, and survival in order to help future research.

## 1. Introduction

Aquaculture is one of the fastest-growing food sectors due to the high population demand for food and the decrease in natural fish stocks [1]. This industry contributes 52% of fish for human consumption and 46% of the total livestock production [2]. Sea bass production in Europe is estimated at 309,226 tons in 2022, and sea bass is one of the most important aquaculture species in Mediterranean countries, especially in Turkey, Greece, Egypt, and Spain [3]. The production of European sea bass is carried out in almost all countries of the Mediterranean. During their first month of life, larvae feed on brine shrimp and rotifers. Afterwards, they begin to consume feed. There are different production methods: floating nurseries at sea, concrete tanks, or ponds on land. Commercial sizes range from 250 g to more than 2500 g. Normally, it takes between 20 and 24 months to reach 400 g from the time the larvae hatch from eggs [3].

Nowadays, aquaculture tends to increase the amount of production to satisfy the food and animal protein human demand through high fish-stock density [4]. To meet this demand, industrial and high-scale aquaculture has to solve many gaps. Overcrowding gives rise to the appearance of diseases due to the stress conditions that fish livestock experience [5]. The main diseases in aquaculture farms are produced by bacteria, which cause great economic losses [6,7]. Bacterial infections dominate the disease reports of European sea bass in the Mediterranean (75%). Reports confirmed *Vibrio* spp., *Photobacteria* spp., and *Tenacibacillus* spp. as the most frequent pathogens in European sea bass [8]. In many cases, antibiotic treatment is beyond the reach of environmental and public health constraints. The administered antibiotics are absorbed at a certain rate, and the unabsorbed treatments go into the environment [9,10] and could promote antibiotic-resistant bacteria [11,12]. Multidrug-resistant bacteria are one of the greatest challenges in public health [13,14]. 

For this reason, researchers have been looking into new alternative approaches such as probiotics. Probiotics, which comes from the Greek terms *pro* and *bios*, are “live micro-organisms which when administered in adequate amounts confer a health benefit to the host” [15,16]. Based on this definition, we considered probiotics as live microalgae, live yeasts, and live bacteria that provide benefits to the host. The use of probiotics in aquaculture production is presented as an ecological, safe, and viable alternative to antibiotics [17]. Moreover, the correct and effective use of probiotics can avoid great economic losses; although their production has certain costs at an industrial level, their application can generate economic benefits [18]. 

The application of probiotic components on fish causes interactions with host intestinal bacteria. These interactions lead to the formation of a wide variety of metabolites, which could produce beneficial outcomes for the fish [19]. Probiotics enhance host parameters such as growth or nutrient assimilation, immunomodulation, disease resistance, and survival rates and mitigate environmental stress [20]. In addition, probiotics can modify the association between the host and microbe or even the microbial community. They also improve the utilization of feed by increasing its nutritional value and enhancing the host’s immune response against different pathogens. Commonly, the application of probiotics in fish industries has been administered via water or feed additives, either singly or in combination with other products or vectors [21,22]. 

Thus, probiotics have been tested in aquaculture with diverse and interesting results. 

Therefore, the aim of this review is to emphasize probiotics’ effect and current role on European Sea Bass aquaculture and provide key findings to promote future research.

## 2. Probiotics Sources and Selection Criteria

### 2.1. Probiotics Sources

Microbes are generally found naturally in humans, animals, soil, sediment, snow, and fresh, brackish, and salt water [23]. Numerous microorganisms have been used in aquaculture due to their probiotic qualities [24]. Normally, these microorganisms are found in fish gastrointestinal tracts, and, through several selection methods, they are isolated and cultivated for use as a probiotic [25]. *Bacillus* spp. is one of the most frequently used probiotics in aquaculture. This frequency is likely due to its ability to sporulate forming endospores, which increases the survival capacity in the gastric tract by resisting exposure to gastric acid, and to its dual aerobic and facultative anaerobic nature, which explains why it can grow in numerous environments [26,27,28,29]. The most common probiotics in European sea bass in recent years are bacteria, specifically *Bacillus* spp., *Pediococcus* spp., *Lactobacillus* spp., *Vibrio* spp., *Shewanella* spp., and *Vagococcus* spp. [30,31,32,33,34,35,36,37,38,39,40,41,42,43,44,45,46,47,48,49,50,51,52,53]. This commonality stands in contrast to the scarce existing bibliography on live yeast and microalgae probiotics in European sea bass [54,55,56,57].

### 2.2. Selection Criteria for Probiotics

Numerous authors have described the necessary characteristics to qualify a microorganism as a probiotic. Necessary requirements for a probiotic to be effective and qualified as such are listed as follows [23,24,25,58,59,60]:(a)The microorganism should be able to adhere to and grow in the host. Then, it should be able to tolerate the bile, gastric juice, and host pH.(b)The probiotic candidate must be free of antibiotic-resistant genes and must not modify heritable traits of the host organism.(c)The microbe should benefit the host system by enhancing the growth or/and development of the immune system against pathogens. It also should have antimicrobial properties.(d)The probiotic candidate should not have harmful effects on the host.

The evaluation of probiotics is carried out through in vitro or/and in vivo tests. In fact, many assays can be carried out both in vitro and in vivo.

The in vitro evaluation should analyze resistance to bile and pH, adherence factors, anti-pathogenic effect, and non-antibiotic resistance.

On the other hand, the in vivo evaluation of the probiotic candidate must show beneficial effects in the host (increasing the immune response, growth and absorption and utilization of food, modulation of intestinal microbiota, and reducing stress), not have harmful effects—assessed by using a biosafety assay—and improve the diseases resistance with an experimental challenge (see Figure 1).

## 3. Technological Aspects and Administration Routes of Probiotics

Technological aspects for the production of probiotics must be considered, as their manufacture and storage can affect the stability of the microorganism. The probiotics that are administered through food must be able to withstand processes of pH, temperature, and pressure [25]. Probiotics are generally supplied frozen or dried, either as freeze-dried or spray-dried powders, and encapsulated [61]. Probiotic delivery methods are diverse and often depend on the type of facility, age, and species of fish [62]. Currently, the methods of administration in aquaculture are injection or addition to the water column or feed [24,63]. Certain factors must be taken into account before choosing the route of administration. The injection generates stress for the fish, and it is complicated and expensive in fish in the larval stage [64]. The advantage of this technique is the guarantee that the fish receives the desired dose of the probiotic. On the other hand, the direct addition of probiotics to the water column could be applicable to all stages of fish [64]. Feed administration is one of the simplest methods, although dry food is contraindicated in larval stages due to the size of the larval mouth [62]. Regarding the investigation of European sea bass, the most common routes of administration are dry food [31,36,39,41,47,48,51,52,53,54,55,56], vectors [30,32,33,34,37,48,57,65], and addition to the water column [35,38,40,42,43,44,45,46,48,57].

## 4. Probiotic Modes of Action in European Sea Bass

Probiotics are an effective prophylactic treatment against different diseases in fish. Determining the mechanism of action by which a probiotic benefits the host is complex. The synergy between various modes of action and/or the interaction with different microbes may result in host benefit [59]. In fact, some authors disagree on the correlation between in vitro and in vivo results. Tinh et al. [66] elaborate an interesting review of the mechanisms of action such as colonization of the gut epithelium, production of inhibitory substances, competition for chemicals or available energy, nutritional contribution, green-water effect, interference with quorum sensing, and immunostimulatory function. Based on the large number of mechanisms that a probiotic can use to exert its action, to date, there is no complete agreement on the results obtained in vivo. Therefore, an increase in research is recommended by the research community to reinforce knowledge of how probiotics work [66,67]. Among the several mechanisms used by probiotics in different microorganisms on European sea bass, the most common are the modulation of immune parameters, competitive exclusion for adhesion sites, production of inhibitory substances, and nutrient competition—digestion and enzymatic contribution (see Figure 2).

### 4.1. Modulation of Immune Parameters

The modulation of immune parameters by probiotic bacteria is diverse and complex. The immune system responds to pathogen-associated molecular patterns (PAMPs) present in pathogens. Pattern recognition receptors (PRRs), fundamental in the innate response, attract pathogens and bind to their PAMPs, triggering the activation of the innate immune response. The best-known PRRs are toll-like receptors (TLRs), which are transmembrane proteins expressed in different immune and non-immune cells [68], one of which is toll-like receptor 2 (TLR2). Moreover, researchers have argued that probiotics possess microbe-associated molecular patterns (MAMPs) able to be detected by the host’s PRRs, triggering, after detection and binding, an intracellular signaling cascade leading to the expression of effector molecules such as cytokines [69]. TLR2 has the capacity to recognize peptidoglycan, which is a main component of Gram-positive bacteria’s cell walls, including lactic acid bacteria (LAB) probiotics [70]. TLR2 stimulation enhances the production of proinflammatory cytokines, such as IL-1β and TNF-α, and induces nitric oxide (NO) synthase. Also, TLR2 stimulation promotes the production of reactive oxygen species (ROS) and nitrogen species, essentials for mechanisms related to host antimicrobial defense. In addition, TLR2 activation has a crucial role in transepithelial resistance against pathogen bacteria [71,72]. Thus, these operations enhance a host’s innate immune system in myriad ways such as increasing the production of lysozymes; enhancing phagocytosis and respiratory burst activity; and enhancing complement activity, peroxidase, antiprotease activity, and cytokine production [2,73]. Moreover, some probiotic components contain specific receptors promoting the production of white blood cells (WBCs) [74]. As proof of this immunomodulation in European sea bass, the following results are collected and detailed in Table 1 [33,34,36,38,40,41,44,45,47,50].

### 4.2. Competitive Exclusion for Adhesion Sites

Bacterial adhesion to host tissues is one of the mechanisms that pathogenic bacteria use to establish their infections [75]. The action of probiotics, on many occasions, is to prevent this adhesion of pathogens, and this action can be specific due to the adhesion of probiotics to the pathogen or to its receptor molecules in epithelial cells or non-specific due to the presence of physicochemical agents [17]. Passive and steric forces, lipoteichoic acids, electrostatic interactions, and specific structures such as external appendages covered by lectins can make this adhesion possible [76]. Bacteria tend to compete with each other by the exclusion of or reduction in other species’ growth. The exclusion of adhesion sites is the main result of several mechanisms and properties of probiotic bacteria to suppress pathogen adhesion [77]. This competitive exclusion of adhesion sites inhibits the action of pathogenic bacteria by blocking infection pathways [78]. In fact, this ability to compete for the binding site with a pathogen is considered one of the main identification criteria for a probiotic [59,76,79,80]. The interaction between surface proteins, produced by certain probiotic bacteria, and mucins creates specific properties that may inhibit the adhesion of pathogenic bacteria [81]. Regarding European sea bass, the adhesion of probiotics (*Vagococcus fluvialis* and *Bacillus velezensis*) in intestinal mucus showed excellent results compared to a control [36,49].

### 4.3. Production of Inhibitory Substances

The production of inhibitory substances is presented as an absolute advantage of probiotics [82]. There is a wide range of inhibitory substances produced by probiotics. Siderophores, lysozymes, hydrogen peroxides, proteases, and antibacterial peptides—including organic acids, antimicrobial peptides, and bacteriocins—are all responsible for pathogen inhibition [23,67,76]. The organic acids produced by LAB, mainly acetic acid and lactic acid, have the ability to penetrate pathogenic bacteria, reducing their intracellular pH or accumulating and causing the death of the pathogen. Therefore, they are considered the main probiotic antimicrobials against Gram-negative bacteria [83]. In addition, two methods of bacteriocins-mediated pathogen clearance have been demonstrated: one includes cell wall perforation, and the other uses inhibition of cell wall synthesis [84]. Regarding antimicrobial peptides, dicentracin is an antimicrobial peptide exclusively produced by European sea bass. Dicentracin has the ability to lysis a wide range of different pathogens, bacteria being the most known [50,85]. The production of antimicrobial substances is not only directed against the lysis of the pathogen but also may be aimed at modifying the environment to make it less suitable for its competitors [2,86]. Makridis et al. [65] used *Phaeobacter* sp. to improve the rearing of European sea bass larvae, showing an in vitro inhibitory effect against *Vibrio anguillarum*. *Bacillus subtilis* was tested in vitro against vibriosis in European sea bass larvae. Its supernatants presented a significant reduction in pathogen growth [37]. In addition, previous research demonstrated the in vitro antagonistic capacity of *Vibrio lentus* as a probiotic against six sea bass pathogens without pathogenic effects on European sea bass larvae [42]. These facts might be attributed to the production of bacteriocins by probiotics. The same results were obtained by Öztürk and Esendal [48], namely that the presence of *Lactobacillus rhamnosus* through *Artemia nauplii* considerably decreased *Vibrio* spp. in European sea bass cultures. Additionally, El-Sayed et al. [57] demonstrated the antibacterial effects of different probiotic microalgae in water against pathogenic bacteria. On the other hand, Monzón-Atienza et al. [50] showed that the dietary administration of *B. velezensis* D-18 enhanced the dicentracin gene expression. Also, Guardiola et al. [40] showed different modifications of antimicrobial peptide gene expressions after *Shewanella putrefaciens* Pdp11 supplementation. 

### 4.4. Nutrient Competition: Digestion and Enzymatic Contribution

Nutrients are essential for bacterial growth. The use of similar nutrients gives rise to hostile competition among species [87,88]. The utilization of available nutrients in environments by probiotics restricts their use by pathogenic microbes [75,77]. In fact, this restriction resulting from competition for nutrients is one of the main mechanisms used by probiotics to inhibit pathogens [23,89]. Iron is one of the most important nutrients for pathogenic bacteria since it is related not only to growth but also to virulence [90,91]. For instance, *Bacillus* spp. has shown a capacity to synthase siderophores and also has a higher organic carbon utilization [92,93]. The absence of iron and carbon limits microbes’ pathogenic functions. Furthermore, probiotics have the capacity to release a wide range of digestive enzymes. Thus, an increase in digestive enzymes can lead to the degradation of nutrients [94]. This digestive enzyme action can increase host nutrient absorption [95]. Both probiotic actions limit the use of nutrients by pathogenic bacteria. 

Several probiotics have been tested in European sea bass and have been observed to enhance the production of enzymes. For one, after the application of *Virgibacillus proomii* and *Bacillus mojavensis*, phosphatase alkaline and amylase presented higher values [43]. Also, the simultaneous administration of *Lactobacillus farciminis* and *Lactobacillus rhamnosus* over 86 days upregulated acid phosphatase activity at day 8 and downregulated acid phosphatase activity at day 23 and a-amylase activity at days 8 and 103 post-administration. Furthermore, trypsin activity presented an increase from days 8 to 103 [31]. In reference to yeasts, various studies by Tovar-Ramírez et al. [54,55] demonstrated the enzymatic modulation capacity of these probiotics in European sea bass. On the other hand, some authors have shown that the application of *Bacillus amyloliquefaciens* for 42 days is capable of modifying the bacterial intestinal flora in European sea bass and reducing the presence of pathogens, surely due to competition for nutrients [52]. 

In recent years, the study of how probiotics are related to the antioxidant response that occurs in the hosts has had a very important boom, carrying out studies to modulate the redox status of the host via their metal ion chelating ability, antioxidant systems, regulating signaling pathways, enzyme-producing ROS, and intestinal microbiota. The mechanisms of how they act are still not fully understood, and future studies are required to clarify the action of probiotics on the antioxidant response of the hosts [96].

## 5. Probiotic Benefits in European Sea Bass Aquaculture

### 5.1. Increased Growth and Survival Rates

Probiotics in aquaculture promote fish growth by improving feed-conversion rates. The survival rate is another parameter that benefits after probiotic implementation [97]. As summarized in Table 1, the application of different probiotics (single or combination) on European sea bass has been reported to promote growth, growth performance, and survival [30,31,32,36,37,39,42,43,45,47,48,49,50,51,53,54,55,56,57,65].

### 5.2. Disease Resistance and Health Status

Like other species, European sea bass are susceptible to pathogen bacteria, viruses, fungi, and parasites [98,99,100]. The application of probiotics in European sea bass has been shown to provide disease resistance. For instance, the administration of *Bacillus velezensis* D-18 at 10^6^ CFU/g over 30 days in European sea bass increased survival against *Vibrio anguillarum* [50]. *Bacillus velezensis* also increased the cumulative survival rates against *Vibrio harvey* SB [42]. Similarly, the supplementation of *Phaeobacter* sp. at 5 × 10^7^ CFU/g in European sea bass fed via diets for 60 days increased resistance against *V. harveyi* [65]. Sorroza et al. [36] found a high survival rate against *Vibrio anguillarum* after the application of *Vagococcus fluvialis* at a high concentration (10^9^ CFU/g) when compared with a control group. Likewise, both probiotic *Bacillus subtilis* and *Lactobacillus plantarum* at 10^6^ CFU/mL demonstrated an increase in disease resistance in European sea bass against *Vibrio anguillarum* [37]. In addition, the presence of *Pediococcus acidilactici* in European sea bass increased survival against *Vibrio anguillarum* [47].

In relation to the health status of the European sea bass after the administration of probiotics, different responses are affected, such as stress modulation, antioxidant status, hematological values, malformations, and parameters of the aquatic environment. Regarding stress, Lamari et al. [41] showed the capacity of *Pediococcus acidilactici* to downregulate HSP70 at 41 days post-hatching in European sea bass larvae. The HSP70 overexpression gene is considered a sign of improvement in acute stress resistance [101]. Silvi et al. [32] tested the effects of *Lactobacillus delbrueckii subsp. delbrueckii* and found a stress decrease in treated European sea bass larvae. The same results were obtained by Carnevali et al. [30] after the administration of *Lactobacillus delbrueckii subsp. delbrueckii* in European sea bass, showing a decrease in cortisol levels. In addition, the application of *Vibrio lentus* at four, six, and eight days post-hatching (dph) in European sea bass larvae had beneficial effects on stress by reducing glucocorticoids [46].

Free radical formation occurs following different processes such as phagocytic activity as well as cellular metabolism [26], which can lead to loss of biological function, tissue damage, and homeostatic imbalance [102]. The formation of free radicals in fish occurs naturally after different metabolic processes [26]. The presence of antioxidant substances is a fundamental factor in the elimination of free radicals. Antioxidants can be divided into enzymatic and non-enzymatic [96]. It is well known that probiotics have the ability to produce enzymes or antioxidant substances or encourage the host to produce them [26]. In fact, several studies have investigated the modification of the oxidative state after probiotic treatment in European sea bass. In one case, the presence of *Shewanella. putrefaciens* Pdp11 in an experimental diet enhanced the oxidative status and the gene expression of superoxide dismutase (SOD) in European sea bass [40]. Salem and Ibrahim [53] also demonstrated that the sole application of *Bacillus subtilis* HS1 decreased the levels of SOD, catalase (CAT), and total antioxidant capacity (TAC) in European sea bass. In contrast, the symbiotic application of that probiotic with chitosan enhanced SOD, CAT, and TAC. Furthermore, not only does the application of probiotic bacteria have these effects, but also the administration of live yeast—*Debaryomyces hansenii* CBS 8339—showed a considerable decrease in antioxidant status [56]. 

Regarding other health status parameters, *Vibrio lentus* enhanced cell proliferation (haematopoiesis), iron transport, and cell adhesion in European sea bass larvae [45]. 

Several authors have described the beneficial effects of probiotics in reducing malformations. In European sea bass, the combination of two different *Bacillus* species—*Lactobacillus farciminis* and *Lactobacillus rhamnosus*—over 86 days at 10^8^ CFU/g in feed considerably reduced malformations [31] as well as the probiotic application of *Lactobacillus rhamnosus* in European sea bass [48]. Additionally, live *Debaryomyces hansenii* reduced malformation appearance in European sea bass larvae [54,55].

On the other hand, the surrounding medium is a fundamental factor in fish wellbeing, so water quality is considered an important parameter [103]. Indeed, Eissa et al. [51] demonstrated that the administration of *Pediococcus acidilactici* in European sea bass culture improved water parameters and led to fish welfare as well as the application of live microalgae on water, which reduced the number of different pathogenic bacteria strains [57]. All of these data are summarized in Table 1.

### 5.3. Elevation of Immune Parameters

The application of probiotics enhances disease resistance by bolstering the immune system as well as general health. It has been demonstrated that probiotics improve different immune parameters in sea bass. In particular, non-specific immune parameters such as lysozyme activity, phagocytic activity, and respiratory burst as well as serum complement activity and the number of macrophages, lymphocytes, erythrocytes, and granulocytes have been modulated after the administration of probiotics in European sea bass [33,34,38,40,44,45,50]. Furthermore, research has shown different modulations in cytokine levels after probiotic supplementation in European sea bass [33,34,38,40,41,42,44,50]. In fish, an increase in immune parameters is usually related to higher survival rates. Several research studies of European sea bass have verified a high survival rate against pathogens after probiotic applications [36,37,42,47,48,49,50,53,65]. All information is summarized in Table 1.

### 5.4. Gut Morphology and Changes in Microbial Diversity

Symbiotic relationships between host and microbes are present in fish. Host and environment—biotic and abiotic factors, respectively—play a fundamental role in intestinal microbiota modulation [104]. Microbes secrete metabolites, producing effects on intestinal environments and triggering changes in host physiology [2]. Probiotics via intestinal–environment interactions may change host intestinal morphologies, thus increasing the surface absorption area localized in the mucosa and microbial diversity [105]. That results in beneficial changes in host metabolism and energy expenditure [106]. Changes in microbial diversity after probiotic supplementation have been related in European sea bass. Through denaturing gradient gel electrophoresis, Makridis et al. [65] demonstrated an increase in bacterial diversity in European sea bass after the application of *Phaebacter* sp. The dietary administration of *Bacillus amyloliquefaciens* spores at 10^7^ CFU/g had implications on gut morphology and microbial diversity in European sea bass. Previously, Silvi et al. [32] showed that the application of *Lactobacillus delbrueckii subsp. delbrueckii* in European sea bass modulated gut microbiota. Moreover, other studies have demonstrated an increase in the number of goblet cells, an increase in the villi length, and the absence of cyst formation, which is a clear indicator of an improvement in gut morphology. Also, after probiotic application, microbial diversity also benefited from a decrease in the Actinobacteria phylum and Nocardia genus. In addition, the number of Betaproteobacteria and Firmicutes—as beneficial bacteria—was higher [52]. All data are summarized in Table 1.

## 6. Highlight Notes for Further Investigation

Although European sea bass are one of the most used species in European aquaculture, especially in the Mediterranean region, they are surprisingly underexplored in research compared to other global species. Species such as tilapia, carp, trout, and even Asian sea bass are well researched in reference to probiotics [2,107,108,109]. Apart from the aforementioned European sea bass references, numerous investigations have been described on the use of probiotics in Atlantic and Mediterranean species such as sole [110,111,112], sea bream [113,114], and turbot [115,116]. However, they are still scarce compared to the global species mentioned above. For instance, the number of microorganisms used as probiotics in Nile tilapia is not nearly comparable to that in European sea bass. This should encourage future research into the framework of this species. Based on the fact that it is a science yet to be investigated, it is possible to delve deeper into probiotic modes of action. Today, it is well known that probiotics have different mechanisms of action as previously described. However, it would be naive to assume that all mechanisms of action are already described. Techniques such as fluorescent in situ hybridization (FISH), different staining methods, and novel microscopy techniques can help to better understand and monitor the behavior of probiotics in hosts and likely identify new mechanisms of action. In fact, the use of European sea bass as a probiotic study model could help to better understand the mechanisms of action in this species. To this end, we recommend the use of germ-free models, as Galindo-Villegas et al. [117] used with zebrafish and Dierckens et al. [118] used with European sea bass, among other studies. Apart from the aforementioned probiotic modes of action in European sea bass, there are other modes that have not been studied in European sea bass such as the inhibition of quorum sensing, also called quorum quenching. Quorum sensing is responsible for several bacterial activities such as biofilm and virulence [119]. However, the literature on quorum quenching by probiotics on European sea bass protection is non-existent. Nonetheless, it is true that quorum quenching of pathogens by probiotics may imply that they can serve as candidates in European sea bass. Other studies have tested it with other aquaculture species such as zebrafish [120] and rainbow trout [121]. The production of inhibitory substances against pathogens is an important probiotic quality [60]. However, studies that describe this production of inhibitory substances by probiotics in European sea bass are scarce. Although the antibacterial activity of probiotics in European sea bass has been published, no reference to antiviral and antifungal probiotic activity has been published yet. The production of these substances by the probiotics, their detection and identification by techniques such as high-performance liquid chromatography (HPLC), and their application in vitro or in vivo in European sea bass may be of great interest to the scientific community.

Sea bass is a species with a very low stress threshold [122]. Chronic stress is one of the main culprits for the immunosuppression of fish in aquaculture farms [123], causing their death. Therefore, the surrounding environment status is a crucial factor. Improving the water quality is another probiotic mechanism of action that confers benefits to the fish, improving the environmental quality [124]. *Bacillus* spp. has the capacity to convert organic matter into CO_2_ and balance phytoplankton production [89]. Certain bacteria are capable of regulating the pH of water in recirculatory aquaculture systems (RASs) by reducing ammonia. The application of novel probiotics in RASs and in biofilters has not been tested in European sea bass. The brief existing literature on water quality improvement after probiotics application in European sea bass comes from Eissa et al. [51] and El-Sayed [57]. However, several studies have demonstrated in other species that the use of probiotics could improve water quality and benefit fish health [23,125]. 

Future and additional studies about mechanisms of action in European sea bass could focus on profiling the transcriptome and proteome of host gut microbiota; the interactions between host, microbe, and gut; the intestinal epithelium; tissues associated with the immune system; antioxidant status; and the antagonistic and synergistic effects of probiotics.

Probiotic effects on a host depend on the duration and dose of administration. Previous research—described in this review—applied an administration period of fewer than 2 months. However, research in other species such as tilapia used longer time periods of up to 8 months [126]. It would be interesting and novel to study the effects on European sea bass of longer administration times.

In reference to the benefits provided by probiotics in European sea bass, there is a variety of information on immunological parameters, survival, growth, and changes in microbiota diversity, previously described. However, there are alternative benefits that have been studied in other species after the administration of probiotics that have been not studied in European sea bass. As noted above, probiotics have the ability to modulate intestinal morphology and microbial diversity. Numerous probiotics have been studied to evaluate their improvement of intestinal morphology and changes in the microbiota. Nevertheless, research on this field in European sea bass is scant, unlike that for other species. In other species, parameters such as the number and morphology of villi, microvilli, lamina propria, and goblet cells have been described by several studies after the application of probiotics [127,128,129,130,131]. Further research could also examine in greater depth the effects between the different probiotic strains applicable to the European sea bass and the host commensal microbiota.

Overcrowding is one of the main factors responsible for chronic stress in fish. However, to date, no studies on the effects of probiotics on European sea bass have been conducted on this topic. Instead, studies on this topic have focused on other species [132,133]. 

Positive changes in blood profiles are also considered an improvement in health status, but, again, few studies on this topic with reference to European sea bass after probiotic effects have been conducted, save for the work of Piccolo et al. [39] and Schaeck et al. [45]. In other species, more blood parameters have been tested such as cortisol, glucose, cholesterol, triglyceride, blood urea nitrogen, bilirubin, plasma total protein, and hematocrit value [134,135]. 

Epithelial surfaces are target areas for possible pathogen invasion [136]. Fish skin abrasions are common injuries in aquaculture, usually due to overcrowded conditions. The skin of the fish acts as a barrier between the host and environment. Additionally, the skin controls homeostasis and provides protection against physical damage [137]. Therefore, the presence of wounds can have a great impact on the economics of aquaculture farms and on animal welfare. Novel research has demonstrated the ability of probiotics to heal wounds [138]. However, no research on this aspect related to European sea bass has been conducted, so these study models could be transferred to European sea bass.

On the other hand, we have been surprised by the few reports we have found regarding the probiotic application of live microalgae or live yeast. Microalgae and yeast have been extended to be used as sustainable feed ingredients for aquaculture. However, the administration of live microalgae or live yeast through vectors—rotifers, *Artemia*, or copepods—in European sea bass larvae could have several beneficial effects not yet described. 

Currently, several probiotic studies could be extrapolated to European sea bass. Thanks to novel techniques that describe bacterial genetic affiliations in the case of probiotic bacteria, new candidate probiotic species are emerging, which may be the object of future research in this understudied species. Nevertheless, when carrying out research with probiotic bacteria both in European sea bass and other species, it would be advisable to deepen the presence of genes with antibiotic resistance, which could be transferred to pathogenic bacteria, still under study. Despite this, the current science remains that probiotics generally have a very beneficial effect on European sea bass, but future research will be needed to elucidate novel mechanisms of action and additional beneficial effects.

## 7. Conclusions

The use of probiotics in European sea bass promotes sustainable production in order to meet the global food demand. The application of these microorganisms improves growth, survival rates, health status, disease resistance, intestinal morphology, and changes in the diversity of the microbiota. Management of doses and duration of administration are essential for the significance of the treatment. Moreover, since the mechanisms of probiotics in aquaculture are not fully understood, the use of probiotics in European sea bass has much room for further study. Investigating the mechanisms of action of probiotics and the effects they produce in European sea bass can provide an invaluable source of knowledge on this species, which, today, is one of the main components of Atlantic and Mediterranean aquaculture.

## Figures and Tables

**Figure 1 animals-13-02369-f001:**
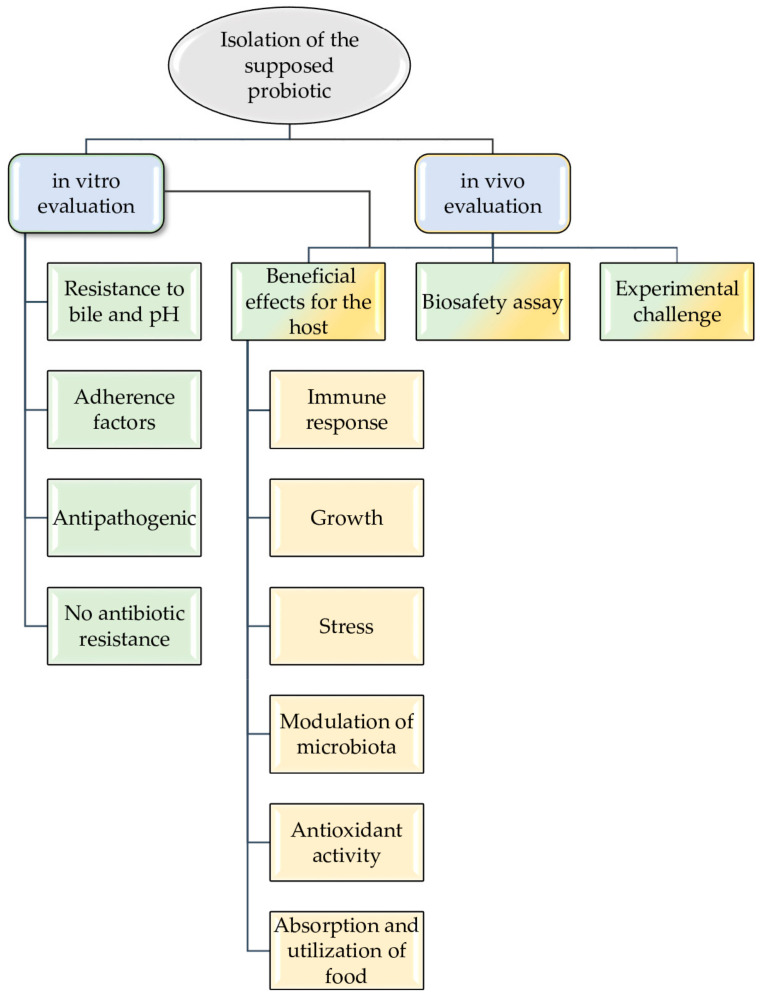
Probiotics selection flow-chart as biocontrol agents in aquaculture.

**Figure 2 animals-13-02369-f002:**
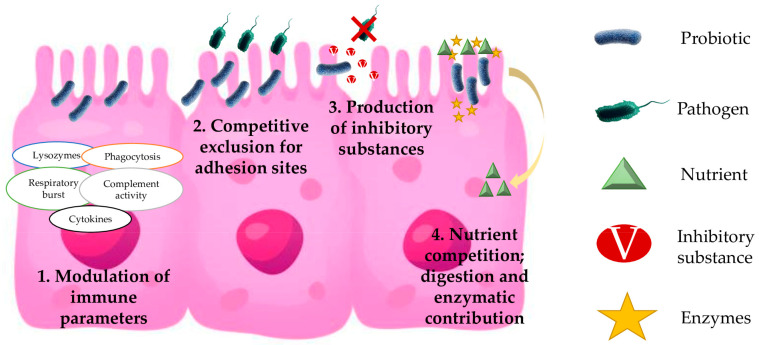
Mechanisms of action of probiotics in European seabass. (1) Modulation of immune parameters—Host immune system responds to microbe-associated molecular patterns (MAMPs) present in probiotics, leading to different intracellular signaling cascades. (2) Competitive exclusion for adhesion sites—Inhibition of pathogen by the colonization of host tissues. (3) Production of inhibitory substances—Production of substances with inhibitory effects on pathogens by probiotics. (4) Nutrient competition (digestion and enzymatic contribution)—Use of nutrients by probiotics, preventing their use by pathogens. Modulation of digestive enzymes that could increase nutrient absorption and improve digestion. Production of beneficial enzymes for the host.

**Table 1 animals-13-02369-t001:** Effect of probiotics on survival, growth, growth performance, immunity, survival against diseases, enzyme production, gut morphology, microbiota, and other parameters in European seabass. (↑) upregulation/increase, (↓) downregulation/decrease. *TcR-β* T cell receptor β-selection*, IL-1β* interleukin beta, *IL-10* interleukin 10, *COX-2* cyclooxygenase 2, *TGF-β* transforming growth factor beta, *Mx* myxovirus resistance proteins, *CAT* catalase, *HSP70* 70-kilodalton heat shock protein, *TNF*α tumor necrosis factor alpha, *IFN* interferon, *DIC* dicentracin, *fbl* fucose-binding, *SOD* superoxide dismutase, *hep* hepcidine, *rbl* rhamnose-binding, *MHCI-α* major histocompatibility complex class I alpha, *MHCII-β* major histocompatibility complex class II beta, *CD4* cluster of differentiation 4, *CD8-α* cluster of differentiation 8 alpha, *TAC* total antioxidant capacity, *GPX* glutathione, Live (L), heat inactivate (H), UV-Light inactivate (UV), only probiotic bacteria (B), high prebiotic level plus probiotic (HPB), and low prebiotic level plus probiotic (LPB).

**Probiotic Bacteria**	**Doses of Administration and Duration**	**Observations**	**References**
***Lactobacillus delbrueckii* subsp. *delbrueckii* **	10^5^ bacteria/mL **Long treatment:** From 11 to 29 days post-hatching: via *Brachionus plicatilis* From 30 to 70 days post-hatching: via *Artemia nauplii* **Short treatment:** From 30 to 70 days post-hatching: via *Artemia solely*	(↑) Growth performance (↑) Body weight (↓) Cortisol	[30]
***Lactobacillus farciminis* CNCM MA27/6R** **+** ***Lactobacillus rhamnosus* CNCM MA27/6B**	10^8^ CFU/g 86 days	(↑) Survival rates (↓) Malformations (↑) Acid phosphatase activity (8 day), trypsin activity (↓) Acid phosphatase activity (23 day), α-amylase activity	[31]
***Lactobacillus delbrueckii* subsp. *delbrueckii* **	10^5^ bacteria/mL **Early treatment:** From 11 to 29 days post-hatching: via *Brachionus plicatilis* From 30 to 70 days post-hatching: via *Artemia solely* **Later treatment:** From 30 to 70 days post-hatching: via *Artemia solely*	Modify gut microbiota (↑) Survival (↓) Stress (cortisol)	[32]
** *Lactobacillus delbrueckii* **	10^5^ bacteria/cm^3^ From 11–29 days post-hatching: via *Brachionus plicatilis* From 30–74 days post-hatching: via *Artemia**nauplii*	(↑) T cells (↑) Acidophilic granulocytes (↑) TcR-β gene expression (↓) L-1β, IL-10, COX-2, and TGF-β gene expression	[33]
** *Lactobacillus delbrueckii* **	10^5^ bacteria/mL From 11–29 days post-hatching: via *Brachionus plicatilis* From 30–74 days post-hatching: via *Artemia salina*	(↑) T cells (↑) Acidophilic granulocytes (↑) TcR-β gene expression (↓) L-1β, IL-10, COX-2, and TGF-β gene expression	[34]
** *Vagococcus fluvialis* **	10^6^, 10^7^, and 10^8^ CFU/mL (in vitro) 30 min incubation	10^8^ CFU/mL as best results: (↑) phagocytosis (10^8^ CFU/mL) (UV>L>H) (↑) Peroxidase (10^8^ CFU/mL) (UV>L>H) (↑) Respiratory burst (10^8^ CFU/mL) (UV>L>H)	[35]
** *Vagococcus fluvialis* **	10^9^ CFU/g 20 days	(↑) Survival against *Vibrio anguillarum*	[36]
** *Bacillus subtilis* **	7 × 10^9^ CFU/mL For 5 days: via *Artemia nauplii*	(↑) Survival against *Vibrio anguillarum*	[37]
***Vagococcus fluvialis* L-21**	10^8^ CFU/mL (in vitro) 1 h incubation	**Mx gene expression:** (↑) 12 h (H), 48 h (L)(H)(UV) (↓) 1 h (L)(H)(UV), 24 h (L)(H)(UV) **IL-1β gene expression:** (↑) 1 h(L)(H)(UV), 48 h (H). (↓) 12 h (L)(H)(UV), 24 h (L)(H)(UV) **IL-6 gene expression:** (↑) 1 h (L), 24 h (H), 48 h (L)(H)(UV) (↓) 12 h (L)(H)(UV) **TNF-α gene expression:** (↑) 1 h (L)(H)(UV) (↓) 12 h (L)(H)(UV), 24 h (L)(H)(UV), 48 h (L)(H)(UV) **IL-10 gene expression:** (↑) 1 h (L)(H)(UV), 12 h (UV), 48 h (L) (↓) 24 h (L)(H)(UV) **COX-2 gene expression:** (↑)1 h (L)(H)(UV), 12 h (L)(H), 24 h (L)(H), 48 h (L)(H)(UV)	[38]
** *Lactobacillus plantarum* **	10 × 10^9^ CFU/kg 90 days	(↑) Survival (↑) Blood cholesterol and triglycerides	[39]
***Lactobacillus casei* X2** ** *Pediococcus acidilactici* **	10^7^ CFU/g 40 days	***Lactobacillus casei* X2** (↑) IL-1β gene expression (↑) CAT gene expression (↓) HSP70 gene expression ***Pediococcus acidilactici*** (↑) IL-1β gene expression (↓) CAT gene expression (↑) HSP70 gene expression	[41]
** *Vibrio lentus* **	10^6^ CFU/mL At 4, 6, and 8 days post-hatching	(↑) Disease resistance against *V. harveyi* SB	[42]
** *Virgibacillus proomii* ** **+** ** *Bacillus mojavensis* **	10^6^ CFU/mL 60 days	(↑) Growth performance (↑) Phosphatase alkaline, amylase activity (↑) Survival	[43]
***Pseudoalteromonas* sp.** ***Alteromonas* sp.** ** *Enterovibrio coralii* ** ** *Lactobacillus casei* **	10^7^ cells/mL (in vitro)	***Pseudoalteromonas* sp.** (↑) Mx (3 h), TNF-α (3 h), IL-10 (3 h) gene expression (↓) Mx (12 h), Caspase 3 gene expression (↓) Lysozyme (1–3 h) (↑) Phagocytosis (↑) Respiratory burst ***Alteromonas* sp.** (↓) Lysozyme (1–3 h) (↓) Mx (3–12 h), Caspase 3 gene expression ***Enterovibrio coralii*** (↑) Mx (3–12 h), IL-10 (3 h) gene expression (↓) Caspase 3 gene expression (↑) Respiratory burst ***Lactobacillus casei*** (↓) Mx (1 h), Caspase 3 gene expression (↑) Phagocytosis (↑) Respiratory burst	[44]
** *Vibrio lentus* **	10^6^ CFU/mL At 4, 6, and 8 days post-hatching	(↑) cell proliferation: hematopoiesis, cell death, ROS metabolism, iron transport, and cell adhesion. (↑) Immunomodulatory functions: pathogen recognition, cytokines, chemokines and receptors, humoral and cellular effectors, IFN-mediated response, and cell death	[45]
** *Vibrio lentus* **	10^6^ CFU/mL 4, 6, and 8 days post-hatching	(↓) Stress	[46]
** *Lactobacillus rhamnosus* **	10^6^ CFU/mL—Rearing Water or 10^8^ CFU/mL From 9 to 50 days post-hatching: via *Artemia nauplii* 10^9^ CFU/g From 50 to 125 days post-hatching	(↓) Deformation (↑) Survival rates (↓) *Vibrio* spp. (after probiotic Artemia)	[48]
***Bacillus velezensis* D-18**	10^6^ CFU/g 20 days	(↑) Survival against *V. anguillarum* 507	[49]
***Bacillus velezensis* D-18**	1 × 10^6^ CFU/g 30 days	(↑) Serum killing percentages (↑) Phagocytic activity (↑) Lysozyme activity (↑) Nitric oxide (↑) IL-1β, TNF-α, and COX-2 gene expression (↑) DIC gene expression (↑) Survival against *V. anguillarum* 507	[50]
** *Pediococcus acidilactici* **	10^10^ CFU/g (2, 2.5, and 3 g) 60 days	(↑) Water quality (↑) Growth performance (↑) Body composition	[51]
** *Bacillus amyloliquefaciens* **	10^7^ CFU/g 42 days	(↑) Villi length (↑) Goblet cells number (↓) Cyst formation (↓) Actinobacteria phylum and Nocardia genus (↑) Betaproteobacteria and Firmicutes	[52]
***Phaeobacter* sp.**	5 × 10^7^ bacteria/mL From 8 to 14 days post-hatching: via *Brachionus* sp. From 14 to 32 days post-hatching: via *Artemia metanauplii*	(↑) Survival against *Vibrio harveyi*	[64]
**Probiotic Bacteria Combinate with Prebiotics**	**Doses of Administration and Duration**	**Observations**	**References**
***Shewanella putrefaciens* Pdp11** **+** **Date palm fruits extracts**	10^9^ CFU/mL 2 and 4 weeks	***Shewanella putrefaciens* Pdp11:** (↑) Antioxidant potential (2 and 4 weeks) (↓) Respiratory burst (4 weeks) (↑) Phagocytic capacity (2 and 4 weeks) **Head-kidney gene expression:** (↑) fbl (4 weeks) (↑) IL-1β (2 weeks) (↑) hep (2 and 4 weeks) **Gut gene expression:** (↑) SOD (4 weeks) (↑) hep (2 weeks) (↑) Lysozyme (2 weeks) (↓) hep (4 weeks) (↓) rbl (2 weeks) ***Shewanella putrefaciens* Pdp11 + date palm fruits extracts:** (↑) Antioxidant potential (2 and 4 weeks) (↓) Serum antiprotease activity (2 weeks) (↓) Natural hemolytic complement (4 weeks) (↓) Respiratory burst (4 weeks) (↑) Phagocytic ability (4 weeks) (↑) Phagocytic capacity (4 weeks) **Head-kidney gene expression:** (↑) rbl (2 and 4 weeks) (↑) IL-1β (2 and 4 weeks) (↑) SOD (2 weeks) (↑) hep (2 and 4 weeks) **Gut gene expression:** (↓) rbl (2 weeks) (↓) hep (4 weeks)	[40]
***Pediococcus acidilactici* (Bactocell^®^)** **+** **Mannanoligosaccharides (MOS)**	MOS(%)/BAC: 0/+ (B) 0.6/+ (HPB) 0.3/+ (LPB) 0/0 Control 90 days	**B:** (↑) TNF-α, IL-1β, COX-2, and IL-10 gene expression (↓) MHCI-α, MHCII-β, CD4, CD8-α, and TCR-β gene expression **HPB:** (↑) TNF-α, COX-2, CD4, and CD8-α gene expression (↓) IL-1β, IL-10, MHCI-α, MHCII, and TCR-β gene expression **LPB:** (↑) TNF-α and IL-1β gene expression (↓) COX-2, IL-10, MHCI I-α, and TCR-β gene expression (↑) Survival against *V. anguillarum* 507	[47]
***Bacillus subtilis* HS1** ***Bacillus subtilis* HS1+** **Chitosan**	10^7^ CFU/g From 30 to 45 days post-hatching	**Probiotic:** (↑) Length, weight (↑) Survival (↑) Aspartate aminotransferase specific activity (↓) ALT (↓) SOD, CAT, and TAC **Symbiotic:** (↑) Length, weight (↑) Survival (↑) SOD, CAT, and TAC (↑) Alkaline phosphatase, acid phosphatase enzymes, and total and specific activities	[53]
**Probiotic Yeast**	**Doses of Administration and Duration**	**Observations**	**References**
***Debaryomyces hansenii* HF1** ***Saccharomyces cerevisiae* X2180**	7 × 10^5^ CFU/g From 10 to 42 days post-hatching	***Debaryomyces hansenii* HF1** **At 27 days post-hatching:** (↑) Amylase (↑) Aminopeptidase N, maltase, and alkaline phosphatase **At 42 days post-hatching:** (↑) Survival (↓) Weight, growth (↓) Malformations ***Saccharomyces cerevisiae* X2180** **At 27 days post-hatching:** (↓) Amylase, trypsin (↓) Aminopeptidase N, maltase, and alkaline phosphatase **At 42 days post-hatching:** (↓) Trypsin (↓) Weight	[54]
***Debaryomyces hansenii* CBS 8339**	10^6^ or 6 × 10^6^ CFU/g From 5 to 37 days post-hatching	**10^6^ CFU/g** (↑) Survival (↑) Weight/growth (↓) Malformations **At 26 days post-hatching:** (↑) Trypsin activity, lipase activity, and amylase activity (↑) Aminopeptidase N, maltase, and alkaline phosphatase **At 36 days post-hatching:** (↑) Trypsin activity and mRNA expression, lipase activity and mRNA expression (↓) Amylase activity and mRNA expression **6 × 10^6^ CFU/g** **At 26 days post-hatching:** (↑) Trypsin activity, lipase activity (↓) Amylase activity (↑) Maltase, alkaline phosphatase **At 36 days post-hatching:** (↑) Trypsin mRNA expression; lipase activity and mRNA expression (↓) Amylase activity and mRNA expression	[55]
***Debaryomyces hansenii* CBS 8339**	43 g/kg From 6 to 48 days post-hatching	(↑) Growth performance (↓) GPX, SOD	[56]
**Probiotic Microalgae**	**Doses of Administration and Duration**	**Observations**	**References**
** *Tetraselmis chuii* ** ** *Nannochloropsis salina* ** ** *Isochrysis galbana* ** ** *Chlorella salina* **	6 weeks: via water and *Artemia metanuplii*	(↓) Bacterial pathogens (↑) Growth performance	[57]

## Data Availability

Not applicable.

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
