# Peer review of "Current Status of Probiotics in European Sea Bass Aquaculture as One Important Mediterranean and Atlantic Commercial Species: A Review"

_animals, 2023, doi:10.3390/ani13142369_

Round 1
Reviewer 1 Report (Previous Reviewer 1)
As per the corrections, it answered my suggestions.
Author Response
Dear reviewer,
Thanks for your opinion, we are delighted to have answered all your suggestions.
Reviewer 2 Report (Previous Reviewer 2)
Lines 53-54 “Adaptively, marine bacteria are able to capture these antibiotic molecules and create resistance against them”. Please reformulate this phrase. It seems that marine bacteria develop resistance on purpose
Lines 97-98, please use commas instead of semicolons
Lines 115-121. Please check upon the English grammar and style, and do not repeat the microosrganism at the beginning of each line.
Fig. 1 in the legend “Briefly diagram for selection of probiotics as biocontrol agents in aquaculture” substitute with “probiotics selection flow-chart as biocontrol agents in aquaculture”
In Fig. 1 please substitute evaluation in vitro and evaluation in vivo with in vitro evaluation and in vivo evaluation respectively.
Lines 127-145 Please rewrite it for grammar clarity. Moreover, in line 144 use are instead of were .
Lines 149-155 please rewrite it for grammar clarity
Line 165 The host
Table 1 is only partially visible. Bibliographic references are missing. What does Mx mean?
Lines 205-207 Please reformulate. It is not clear what you want to say
Lines 217-221 Please reformulate. It is not clear what you want to say
Lines 255-257 Please substitute “Nutrients are essential for bacteria as well as mammals. Bacteria, both probiotic and pathogenic, use nutrients and energy to grow. The use of similar sources between species gives rise to a hostile competition [88,89]” with “Nutrients are essential for bacterial growth. The use of similar nutrients gives rise to a hostile competition among species [88,89]”.
Line 270 phosphatase alkaline is not a digestive enzyme
Lines 271-272 In fact, the application stimulated the precocious maturation of the digestive system [44]. What is this supposed to mean?
Lines 281-288. Please rewrite it
Lines 313-315 Based on the references found in European sea bass after the application of probiotics, we classify the effects of probiotics on health status as the modulation of stress, anti-oxidant status, haematological values, malformations, and water parameters. It is not clear what you want to mean.
Lines 326-328 Free radicals have tropism for polyunsaturated fatty acids present in cell membranes, DNA nucleic acids, and proteins causing loss of biological function, tissue damage, and homeostatic imbalance [103]. Please reformulate, it is not clear what you want to say.
Grammatical review is required.
Author Response
Dear reviewer,
Thanks for your opinion, we are delighted to have answered all your suggestions.
Comments and Suggestions for Authors
Lines 53-54 “Adaptively, marine bacteria are able to capture these antibiotic molecules and create resistance against them”. Please reformulate this phrase. It seems that marine bacteria develop resistance on purpose
The phrase has been changed to “Marine bacteria can contact with these antibiotic molecules and that can create resistance against them.”
Lines 97-98, please use commas instead of semicolons
The semicolons have been changed
Lines 115-121. Please check upon the English grammar and style, and do not repeat the microorganism at the beginning of each line.
The style has been corrected, and we have change microorganisms all as we can
Fig. 1 in the legend “Briefly diagram for selection of probiotics as biocontrol agents in aquaculture” substitute with “probiotics selection flow-chart as biocontrol agents in aquaculture”
The legend has been changed
In Fig. 1 please substitute evaluation in vitro and evaluation in vivo with in vitro evaluation and in vivo evaluation respectively.
It is changed according reviewer advise
Lines 127-145 Please rewrite it for grammar clarity. Moreover, in line 144 use are instead of were .
The text has been clarified
Lines 149-155 please rewrite it for grammar clarity
The text has been clarified
Line 165 The host
It is changed according reviewer advise
Table 1 is only partially visible. Bibliographic references are missing. What does Mx mean?
The Table is complete and the references are numbered, it is possible that it is a format problem, but we see it as correct. Mx is interferon-inducible Mx protein and has been clarified in the table.
Lines 205-207 Please reformulate. It is not clear what you want to say
It is reformulate to clarify the idea
Lines 217-221 Please reformulate. It is not clear what you want to say
It is reformulate to clarify the idea
Lines 255-257 Please substitute “Nutrients are essential for bacteria as well as mammals. Bacteria, both probiotic and pathogenic, use nutrients and energy to grow. The use of similar sources between species gives rise to a hostile competition [88,89]” with “Nutrients are essential for bacterial growth. The use of similar nutrients gives rise to a hostile competition among species [88,89]”.
The change has been done according review comment
Line 270 phosphatase alkaline is not a digestive enzyme
The paragraph has been clarified
Lines 271-272 In fact, the application stimulated the precocious maturation of the digestive system [44]. What is this supposed to mean?
This sentence has been deleted
Lines 281-288. Please rewrite it
The paragraph has been rewrite
Lines 313-315 Based on the references found in European sea bass after the application of probiotics, we classify the effects of probiotics on health status as the modulation of stress, anti-oxidant status, haematological values, malformations, and water parameters. It is not clear what you want to mean.
The paragraph has been clarified
Lines 326-328 Free radicals have tropism for polyunsaturated fatty acids present in cell membranes, DNA nucleic acids, and proteins causing loss of biological function, tissue damage, and homeostatic imbalance [103]. Please reformulate, it is not clear what you want to say.
The paragraph has been rewrite to clarify our idea
Comments on the Quality of English Language
Grammatical review is required.
The paper has been revised for an English proficiency
Reviewer 3 Report (Previous Reviewer 3)
I hope you find the notes more than appropriate, and I congratulate you once again for the effort made by daring to address such a complex issue.

It is essential that it be written in a more summarized way, and with a more concise text.
Author Response
CURRENT STATUS OF PROBIOTICS IN EUROPEAN SEA BASS AQUACULTURE AS THE MAIN MEDITERRANEAN AND ATLANTIC COMMERCIAL SPECIES: A REVIEW
ANIMALS 2494257
Figure 1 looks better but need to be improved in the text. Selection criteria for probiotics could be tested / and invitro in vivo. A, b, c, d: put in the figure in the same order than the text. Please try to explain the in vivo part in the text that is not covered.
IT HAS BEEN CHANGED FOR CORRELATION BETWEEN TEXT AN FIGURE 1
Figure 2:
“Among the several mechanisms used by probiotics in different microorganisms on European sea bass, the most common are modulation of immune parameters (4), competitive exclusion for adhesion sites (1), production of inhibitory substances (3), and nutrient competition and enzymatic contribution (2) (see Figure 2)”. Please use the same order in the text than in the figure
IT HAS BEEN CHANGED FOR CORRELATION BETWEEN TEXT AN FIGURE 2
- Modulation of immune parameters / This is a complex topic, and it's not meant to be a general review, but in my opinion, it covers the other three aspects. For example, the production of lysozyme is related to the abundance of populations, adhesion, or competition between microorganisms. Perhaps I would put point 4 as transversal to the other 3 since it is also a way of measuring the improvement of the immune system. Although I repeat it is a very complex
WE AGREE THAT THE MODULATION OF IMMUNE PARAMETERS IS A COMPLEX ISSUE, AND AS THE REVIEW REFERS TO, IT IS NOT INTENDED TO BE A GENERAL REVIEW, SO WE BELIEVE THAT THE CURRENT FORMAT MORE THAN COVERS THE PREDETERMINED PURPOSE FOR THIS PUBLICATION.
Table 1 could be improved:
Please include references and methods of testing/study probiotics in Table 1 because not all the studies are conducted vs pathogens. Include a column with in vivo /in vitro test, modes of administration.
Try to make it easier to read. This is the core of the work.
Try to improve the observations column: I am more than aware that it is not usual to perform the same techniques or activities. I honestly think that the table would be much better if it were horizontal with a column for biological activity (less text and more graphic).
Think in include columns with cross and arrows. This reference analyzed growth (↑), (↓), not analyzed (0); Or try to organize it by observations.
I know it is difficult but it would improve the comparison. That's a lot of information, as gene expression, think to make a complementary table or separate table with gene expression (by gene).
Try not to use abbreviations in the table that are not explained in the table footer As example, Live [L], heat inactivate [H] or UV-Light inactivate [UV].
THANK YOU VERY MUCH FOR YOUR SUGGESTION, THE AUTHORS CONSIDER THAT THE TABLE IS IN AGREEMENT WITH THE TEXT, WELL CORRELATED AND RESPECTING THE ORDER OF MENTION -IN THIS WAY, THE READER DOES NOT GET LOST-. THE WAY TO ORDER THE TABLE BY MICROORGANISMS, INSTEAD OF BY OBSERVATIONS OR IN VITRO OR IN VIVO TESTS, HAS ALREADY BEEN USED BEFORE (HTTPS://DOI.ORG/10.1007/S12602-021-09852-X).
REGARDING THE OBSERVATIONS, WE HIGHLY VALUE YOUR CONTRIBUTION, BUT THE FACT OF PRESENTING THE TABLE IN A MORE SUMMARIZED WAY PROBABLY DEPRIVES THE READER OF INFORMATION.
OUR OBJECTIVE IS TO PRESENT THE READER WITH THE CORRESPONDING INFORMATION, IN A SUMMARIZED WAY BUT THAT CLEARLY EXPOSES WHAT HAS BEEN DONE IN EACH WORK. PRESENTING IT IN ANOTHER WAY WOULD NOT CORRESPOND TO THE VISION WE HAVE OF THIS WORK.IN ADDITION, ABBREVIATIONS HAVE BEEN ADDED IN THE TABLE FOOTER.
Section by section analysis
- Introduction:
Introduction could be more precise, is about 60 lines, 43 to 92 is so extensive,
Paragraph 1 is fine, but paragraphs 2, 3, 4 and 5 could be shortened. We run the risk of losing the reader in the long text and losing the general sense. (It looks like a direct translation in Spanish to the English version)
As an example, I rewrite of part of the introduction:
Please keep this in mind as a suggestion. The work is difficult to read.
Paragraph 2 and following:
Nowadays, aquaculture tends to increase the amount of production to satisfy the food and animal protein human demand through high fish-stock density [4]. And to meet this demand industrial and high scale aquaculture has to solve many gaps.
Overcrowding, that increase stress condition and diseases [5]. Bacterial infection that dominates disease reports (75%) and cause important economic loses at the Sea Bass aquaculture [6,7,8]; Behavior and dominance problems due to overcrowding can result in aggressive behavior, wounds and wounds becoming infected... and probiotic could improve wounds healing [123] Other citation (new) (Ceballos-Francisco et al., 2017; Chen et al., 2020)
And the antibiotic treatment associated to these diseases are not extent of environmental and public health constraints. The antibiotics administered to animals is absorbed at certain rate and unabsorbed treatments goes to the environment [10, 11], and could promote antibiotic resistant bacteria [11,12,13]. This face to the greatest challenges in public health [14,15]. Alternatives as probiotic to antibiotic can mitigate this environmental and public health risk [citation] making aquaculture more responsible with environment.
Probiotics, which comes from the Greek terms pro and bios, are “live microorganisms which when administered in adequate amounts confer a health benefit to the host” [16,17]. Based on this definition, we considered as probiotic live microalgae, live yeasts, and live bacteria that provide benefits to the host.
Probiotics as sustainable alternative correctly an effectively applied can avert great economic losses, while their production costs is high at an industrial level, their application could generate economic benefits [19]. Enhance host parameters as growth or nutrient assimilation, immunomodulation, disease resistance, survival rates, and mitigate environmental stress [18]. Biological benefits of probiotics are widely studied and produce beneficial outcomes for the fish [20, 21, 22, 23] and other livestock… […] , plants […];
In aquaculture, certain effects are more than studied.... against bacterial challenges (Reyes-Becerril et al., 2017); Enhances immunity (Reyes-Becerril, Guluarte, Ceballos-Francisco, Angulo, & Angeles Esteban, 2017) (Please looks MA Esteban bibliography is wider…)
In aquaculture there are different modes of action and of administration of probiotics via water or feed additives, either singly or in a combination with other products or vectors [24,25] [59, 60, 62, 64]
Therefore, the aim of this review is to emphasize on probiotics’ effect and current role on European Sea Bass aquaculture, and provide key findings to promote future research.
WE APPRECIATE THE AVAILABILITY OF THE REVIEWER, AND WE HAVE USED AS MUCH OF HIS COMMENTS IN THE INTRODUCTION AS WE COULD WITHOUT LOSING THE WAY WE WANT TO GIVE THIS PUBLICATION.
2.- Probiotics: Sources, Selection Criteria, Technological Aspects and Administration Routes, and Modes of Action in European Sea Bass
This section looks too wider……. Consider breaking into parts:
2.- Sources and selection
3.- Technological Aspect of Administration Routes
4.- Tested Modes of Action / Probiotic Modes of Action in Sea Bass
Please give it a more relevance as main epigraph
and so on …
- Probiotic Benefits in European Sea Bass Aquaculture
- Highlight Notes for Further Investigation
- Conclusion
IT HAS BEEN CHANGED ACCORDING THE REVIEWER CONSIDERATIONS
Sections 2.4.1, 2.4.2, 2.4.3 and 2.4.4. Must be organized as commented before (1,2,3,4)… and Figure 2
- Highlight Notes for Further Investigation.
DONE
Excessively long paragraphs without any reference, please try to be specific: As example 395 to
409 Lines, or this same paragraph that has 30 lines… This must be improved.
Lines 447 to 458 the same.
THE CONSIDERATIONS HAVE BEEN TAKEN INTO ACCOUNT AND CITATIONS HAVE BEEN ADDED TO REFORCE THIS SECTION.
At the end of the document, you have some typos and references correction.
General concept comments
The use of probiotics is common in both aquaculture research and the supplementation of commercial diets. Many aquaculture companies incorporate probiotics into their formulated feeds to enhance fish performance, mitigate the negative effects of stressors, and reduce the reliance on antibiotics or other chemical treatments. The use of probiotics in commercial aquafeeds is a growing trend, reflecting the recognition of their potential benefits in promoting fish welfare and sustainable aquaculture practices.
The commercial or industrial aspect of probiotic use is not covered in the article. We understand that it may be of great interest to at least reference it. For example, Lallemand's Bactocell is presented in a dehydrated format, with sucrose. Technological aspect of probiotics is covered by Kiron, 2015. While commercial or industrial formats of probiotics are typically dehydrated or lyophilized and may contain prebiotics, the organisms used in research are commonly utilized in their live form. This applies to bacteria, microalgae, and yeasts as well. There is a lack of greater extension regarding yeasts and microalgae. And it is missed if there is any difference between the stages of development, live yeast and microalgae are commonly used in larval feeding. They have thought about that use / or it has been evaluated as a probiotic. The use of compound feed for larval feeding is increasingly common, at shorter days of age. It would be an aspect to consider.
Indeed, there can be some ambiguity and overlap in the definitions and applications of probiotics, prebiotics, symbiotic, and postbiotics in both research and industrial contexts. While there are general distinctions between these terms, their specific usage and categorization can vary depending on the scientific field, regulatory frameworks, and industry practices.
In brief, probiotics refer to live microorganisms (such as bacteria, yeasts, or microalgae) that, when administered in adequate amounts, confer a health benefit to the host. Prebiotics, on the other hand, are non-digestible substances that selectively promote the growth and activity of beneficial microorganisms in the gut. Symbiotic combine probiotics and prebiotics, aiming to enhance their synergistic effects on gut health and overall well-being. Postbiotics, a relatively newer term, refers to the bioactive compounds or metabolic by-products produced by probiotics that can exert beneficial effects on the host independently of the live microorganisms.
Providing a separate section dedicated to defining probiotics would be beneficial in terms of clarity and organization. By including a dedicated section that explains the definition of probiotics, their characteristics, and their intended health benefits, readers will have a better understanding of the term and its significance within the context of the manuscript which is focused to the use of live cells, but some articles in table 1 also use prebiotics together with probiotics, or Torrecillas et al 2018 (reference 41) use Bactocell from Lallemand. It is possible to manipulate certain conditions to bring about a state where growth may be put in “stand-by mode”, yet the microbe remains alive.
For better and separate of the introduction definition of probiotics in the manuscript please take into account references that are listed with the review. From my point of view, the definition should be expanded, as the referenced articles do, to not only include the use of live organisms, which is not common or feasible in industrial food production or aquaculture farms. I have significant expertise in working with probiotics, prebiotics, and immunology, and I consider the definition provided by Merrifield et al. (2010) to be the most accurate and comprehensive. In the same way Jahangiri & Esteban, 2018.
The selection of probiotics as biocontrol agents in aquaculture (Figure 1) is too brief. A better schematic is shown in (Kiron, 2015), reconsider improving and expanding Figure 1. The in vivo evaluation of probiotics or prebiotics does not always have to be accompanied by a biological challenge against a pathogen. There are many works published by MA. Esteban in this sense.
And the in vitro test could be carried out in another way, since the measurement of the immunological parameters is not necessarily due to the inhibition of the pathogen (Salinas et al., 2006).
There is an excellent review of studies of the probiotic SpPdp11 in Cámara-Ruiz et al., 2020. Some published test methods are listed below, and not only probiotics are tested against pathogens: Improved cellular and humoral immunity; Improved growth performance and stress tolerance; Modulation of gut microbiota; Positive proteomic changes in skin mucus under stress; Higher adaptability to dietary changes in the intestinal microbiota and potential protective effect against oxidative stress; increases the transcription of genes related to antiapoptotic effects and oxidative stress regulation; Improved antioxidant activity mainly in gills and skin; Beneficial effects regarding the negative effects in intestinal histology; depressed expression of pro-inflammatory and increased expression of anti-inflammatory cytokines after wounding.
Salinas et al., 2006, Chen et al., 2020, and Cámara-Ruiz et al., 2020 They should definitely be cited.
Chen et al., 2020, above all, it is novel since it introduces the measure of the probiotic during the healing of a wound. Other works of I. Salinas about mucosal health are too interesting.
Selection, characterization and modes of action:
Probiotic selection criteria are a complex subject, technological aspects are well addressed in Kiron, 2015. And other aspects in Merrifield et al., 2010, differences as essential and favorable factors of probiotic selection would be interesting that be included in the work. As Supplementation form (Merrifield et al., 2010 has a good example of this 3.8.2).
THE COMMERCIAL OR INDUSTRIAL ASPECT OF THE USE OF PROBIOTICS IS NOT COVERED AS IT IS NOT ONE OF THE ITEMS TO BE DEALT WITH IN THIS WORK, BEING OF INTEREST AND AS THE REVIEW SAYS, IT HAS ALREADY BEEN DEALT WITH IN PREVIOUS REVIEWS. REGARDING YEASTS AND MICROALGAE, WE HAVE ADDED SOME INFORMATION FROM WHAT WAS FOUND IN THE BIBLIOGRAPHY SHOWING WHAT HAS BEEN DONE WITH THESE MICROORGANISMS. REGARDING THE REST OF THE SUGGESTIONS, WE BELIEVE THAT BASED ON THE WORK THAT WE WANT TO CARRY OUT, WE HAVE INCLUDED EVERYTHING IN ITS PROPER MEASURE, ADDING THE REFERENCES THAT YOU RECOMMENDED IN THE FIRST REVIEW. WE APPRECIATE THE NEW COMMENTS BUT WE BELIEVE THAT INTENSIFYING THE INFORMATION ON THOSE POINTS IS ALREADY RECENTLY TREATED (CHEN, KIRON, ETC.) BY OTHER AUTHORS DO NOT PROVIDE MORE VALIDITY TO OUR PAPER
Modes/mechanisms of action Figure 2 must be improved: Release of enzymes, antioxidant status, wound healing, etc. As previous Cámara-Ruiz et al., 2020; or as example (El-Saadony et al., 2021): The perspective of nutrition is missing.
“Some of the suggested mechanisms of action of probiotics include (i)
competitive exclusion through the production of inhibitory compounds,
(ii) competition for nutrients, chemicals, or energy, (iii) adhesion site competition, (iv) contribution to digestion, (v) contribution to macro-
and micronutrients, (vi) enhancement of immune response, and (vii)
reduction of virulence through quorum sensing (QS) manipulation [77 – 81].”
NUTRITION AND ENZYMATIC CONTRIBUTION HAVE BEEN PREVIOUSLY ADDED IN THE FIGURE 2. REGARDING, ANTIOXIDANT STATUS WE HAVE CONSIDERED IT AS HOST BENEFICIAL EFFECTS. WE UNDERSTAND THAT THE PRODUCTION OF ANTIOXIDANT ENZYMES OR PRODUCTS COULD BE DIRECTLY PRODUCED BY PROBIOTICS OR AFTER MODULATION OF HOST GENE EXPRESSION (DOI:10.3390/NU9050521). NO CURRENT STUDY DEFINES THIS ASPECT IN EUROPEAN SEABASS. FOR THIS REASON WE CONSIDER IT COUNTERPRODUCTIVE TO ADD IT IN THE FIGURE BECAUSE WE ARE FOCUS ON THE MODES OF ACTION IN EUROPEAN SEABASS. HOWEVER, WE DETAILED THE PERTINENT INFORMATION WHEN WE MENTIONED IT. OTHER STUDIES HAVE ALSO NOT -DIRECT- MENTIONED IT IN FIGURES, AS EXAMPLE: DOI: 10.1159/000342079, DOI: 10.1080/01652176.2016.1172132
ON THE OTHER HAND, EVEN THOUGH IT WOULD BE SO INTERESTING ADD OTHER MECHANISM OF ACTION SUCH AS WOUND HEALING, NO REPORTS HAVE BEEN PREVIOUSLY DESCRIBED IT IN EUROPEAN SEABASS EITHER AS A BENEFIT OR AS A MECHANISM OF ACTION. THEREFORE WE DO NOT CONSIDERED INCLUDING IT, AS OTHERS MECHANISMS. HOWEVER, IT HAS BEEN INCLUDE IN FUTURE RESEARCHES TOPIC.
REGARDING “CONTRIBUTION TO DIGESTION AND CONTRIBUTION TO MACRO- AND MICRONUTRIENTS”, WE CONSIDER THAT IT HAS BEEN CORRECTLY ADDED AS: NUTRIENT COMPETITION; DIGESTION AND ENZYMATIC CONTRIBUTION
These 3 references are included in Table 1, they should be separated from the rest in some way as they also include prebiotics in the test. For a novice researcher it might be confusing.
Reference 34 Shewanella putrefaciens Pdp11 and date palm fruits extracts. Reference 41 Pediococcus acidilactici + Mannanoligosaccharides (MOS) Reference 47 Bacillus subtilis HS1+ Chitosan.
THIS SUGGESTION HAS BEEN PREVIOUSLY TAKE INTO ACCOUNT AND HAVE BEEN IMPLEMENTED. SEE TABLE 1 - PROBIOTIC BACTERIA COMBINATE WITH PREBIOTICS.
The "game" of immunostimulation in fish by means of probiotics depends on many factors, dose- response, cultivated species, type of probiotic plus other additives, stages of development and culture systems, etc. Throughout the article the effect of overcrowding is referenced several times. Please try to reference directly if there is any improvement that corrects this factor by using probiotics (regardless of species).
WE HAVE PREVIOUSLY CONSIDERED THIS SUGGESTION AS FUTURE INVESTIGATION: “OVERCROWDING IS ONE OF THE MAIN FACTORS RESPONSIBLE FOR CHRONIC STRESS IN FISH. HOWEVER, TO DATE, NO STUDIES ON THE EFFECTS OF PROBIOTICS ON EUROPEAN SEA BASS HAVE BEEN CONDUCTED ON THIS TOPIC. INSTEAD, STUDIES ON THIS TOPIC HAVE FOCUSED ON OTHER SPECIES [111].”
Scope:
It would be interesting to expand the scope of the study and compare it with other Mediterranean species. Including some references in this regard could improve the study. Therefore, it would be relevant to reference the works of María de los Ángeles Esteban, Miguel
- Moriñigo, and other researchers who have contributed to the broader field of fish immunology and pre, probiotics, etc. But they work mainly in gilthead sea bream or sole.
It would be interesting to expand the scope of the study and compare it with other non- Mediterranean species as salmon or tilapia from the scope of probiotic used.
FOLLOWING THE REVIEWER'S CONSIDERATIONS, AND UNDER THE PREMISE THAT OUR WORK IS FOCUSED ON SEABASS, WE HAVE TAKEN INTO ACCOUNT THE WORK DONE BY OTHER RESEARCHERS ON THE OTHER 2 MEDITERRANEAN AQUACULTURE SPECIES, INCLUDING REFERENCES ON THE WORK DONE ON PROBIOTICS IN SEABREAM AND SOLE.
Other considerations:
The goal of this reviewer is to ensure the article is published in the best possible form. The limited scope of the experimentation and the need for further studies is well understood, which makes it a limited review much remains to be done in sea bass.
And classify epigraphs from the point of view of cultured specie/probiotic type/effects.
THANKING THE REVIEWER FOR HIS COMMENTS AND INDEED THE LIMITATION OF CURRENT INFORMATION, MAKES THIS REVIEW PROVIDE AN OVERVIEW OF THE CURRENT STATE OF RESEARCH ON EUROPEAN SEABASS WITH PROBIOTICS AND ITS PERSPECTIVES.
Typos: Check I didn't do an in-depth review on this aspect.
Tables.
Table 1: Is not Pdp1,is Pdp11
Shewanella putrefaciens Pdp11 and date palm fruits extracts; Reference 34 Line 206: Pdp1 is Pdp11
IT HAS BEEN CORRECTED
References
Correct reference 16.
Hai, N. V. (2015). The use of probiotics in aquaculture. Journal of Applied Microbiology, 119(4), 917- 935. https://doi.org/10.1111/jam.12886
IT HAS BEEN CORRECTED
Please correct reference 3:
Apromar. (2022). La Acuicultura en España 2022 (p. XX) Asociación Empresarial de Productores de Cultivos Marinos. Available online… Accessed …
IT HAS BEEN CORRECTED
Round 2
Reviewer 2 Report (Previous Reviewer 2)
My comments have been addressed. I have no further requests
This manuscript is a resubmission of an earlier submission. The following is a list of the peer review reports and author responses from that submission.
Round 1
Reviewer 1 Report
A well-prepared paper with a relevant theme to aquaculture, it has a lot of important information related to physiology and health, however, because it is about fish production, there is no reference to the financial cost that the treated technique will impose on the final product.
When the subject is related to production animals, references on the subject of costs and financial viability are necessary.
1. What is the main question addressed by the research? About probiotics in aquaculture, the work presented current references. 2. Do you consider the theme original or relevant in the area? that address a specific gap in the field? Theme is relevant, but the authors did not talk about the cost of production using this technology. 3. What does it add to the subject area compared to others published material? The exposed information is interesting for understanding the theme regarding the physiology in general of the use of probiotics. 4. What specific improvements should authors consider regarding the method? What other controls should be considered? Authors should cite economic information. 5. The evidence is consistent with the evidence and arguments presented and do they address the main question posed? Well crafted work. 6. Are references neutral? Yes. 7. Please include any additional comments about tables and figures. No.
Reviewer 2 Report
The review “Current Status Of Probiotics In European Sea Bass Aquaculture As The Main Mediterranean And Atlantic Commercial Species: A Review”, by Monzón-Atienza et al., presents several problems that prevent its publication.
The title speaks of probiotics and seabass, but many paragraphs (in particular 2.1, 2.2, and 2.3) are very generic and report studies of a general nature. The effects of probiotics in Seabass are reported in table 1, but they are insufficiently commented, just a list of scientific literature results. Therefore, the reading is disjointed and it doesn't flow smoothly.
• What is the main question addressed by the research?
Use of Probiotics in European Sea bass aquaculture: a review
• Do you consider the topic original or relevant in the field?
The topic is relevant to the field, however, there are several reviews on the use of probiotics in aquaculture. The authors should explain the rationale behind the review
Does it address a specific gap in the field?
It doesn't look like that. Surely the review would benefit from the addition of an introduction clarifying what the scope of the review is and what critical issues it would like to address.
• What does it add to the subject area compared with other published material?
Frankly, not much. There are several reviews covering the topic of probiotics in aquaculture such as
https://doi.org/10.3389/fmicb.2018.02429
https://dx.doi.org/10.22161/ijfaf.4.5.1
https://doi.org/10.1186/s41936-020-00190-y
DOI: 10.1016/j.fsi.2021.07.007
https://doi.org/10.1080/01652176.2016.1172132
• What specific improvements should the authors consider regarding the
methodology? What further controls should be considered?
NA
• Are the conclusions consistent with the evidence and arguments presented
and do they address the main question posed?
NA
• Are the references appropriate?
yes
• Please include any additional comments on the tables and figures.
There are two figures and one table. Fig. 1 is very general as well as fig 2. Table 1 is fine
The grammar is poor and difficult to read. Many sentences are difficult to understand because of the grammar. Proofreading by a native speaker is crucial.
Reviewer 3 Report
Please see the attached file. I hope the notes will be useful to improve the work.
